# Does the regulatory quality matter in the relationship between climate finance and inclusive growth in Africa?

Isubalew Daba Ayana *

Department of Economics, Wollega University, Nekemte, Ethiopia

* isubalewd@wollegauniversity.edu.et

## Abstract

This study scrutinizes the effect of climate finance on inclusive growth in Africa. It further explores how regulatory quality dynamics impact the connection between climate finance and Africa's inclusive growth. Grounded on panel macroeconomic data straddling 54 African states from 2013 to 2023 and employing the two-step system generalized method of moments (2SYS-GMM) econometric estimation technique, the subsequent outcomes appeared. First, climate finance encourages inclusive growth in Africa, while the interactive terms of climate finance and regulatory quality have a shrinking effect. Before interacting with the regulatory quality, a 1% in increase in climate finance heightened Africa's inclusive growth by 0.3607% in the long run, while it was accompanying with a 0.1561% upsurge in the inclusive growth in the short run, all other factors remaining constant. Contrarily, the system GMM model publicized that in the long run, a one percent increase in the interactive terms of climate finance and regulatory quality of Africa signposts to a 0.775 percent diminution in inclusive growth, while it marks a 0.753 percent lessening in the short run, on average, and other things remaining constant. The study concludes that the feeble regulatory quality of Africa is harmful in both the long run and the short run. This suggests that negative regulatory quality dynamics completely shrink the positive effect of climate finance on the inclusive growth of Africa over the periods under this study. Further, the anticipated benefits of climate finance in fostering Africa's inclusive growth may persist elusive unless noteworthy progresses are made to Africa's currently existing regulatory frameworks.

## 1. Introduction

Climate financing gained prominence after the 2015 Paris United Nations Framework Convention on Climate Change. In a world where climate change is wreaking havoc, climate financing is currently the uppermost topic on the agenda. Climate finance helps countries reduce their greenhouse gas emissions by promoting renewable

**Data availability statement:** The panel data that underlie the findings of this study are available at the World Development Indicators(WDI) URL: https://databank.worldbank.org/source/world-development-indicators/Type/TABLE/preview/on, and UN Trade and Development (UNCTAD) URL: https://unctadstat.unctad.org/datacentre/dataviewer/US.InclusiveGrowth.

**Funding:** The author(s) received no specific funding for this work.

**Competing interests:** The authors have declared that no competing interests exist.

energy sources like wind and solar. It also helps communities adjust to the consequences of climate change [1,2].

Reducing greenhouse gas emissions, strengthening greenhouse gas sinks, and reducing the vulnerability of human and natural systems to the negative consequences of climate change are the objectives of climate finance. Vulnerabilities include reliance on agriculture, inadequate infrastructure, restricted access to technologies, and debt dependency, which put developing countries under more pressure [3]. More than 750 million people, mostly in South Asia and Africa, do not have access to electricity as of 2021, and 2.6 billion people cook using dangerous techniques. Only 28% of Sub-Saharan Africans in rural regions have access to electricity, compared to 79% in urban areas, indicating exceptionally low rates of electrification in rural areas [4,5].

In order to accomplish climate commitments, climate finance must surpass $4.35 trillion annually by 2030, having reached $1.3 trillion in 2021–2022. Vulnerabilities include reliance on agriculture, inadequate infrastructure, restricted access to technology, and debt dependency, which put developing economies under more pressure [6]. For example, just 3.7% of global climate money flowed to Sub-Saharan Africa in 2021–2022, and slightly more than 2% went to Least Developed Countries [4]. In fiscal year 2024, the World Bank Group provided a record of $42.6 billion in climate finance to support efforts to reduce poverty on a habitable planet by investing in cleaner energy, stronger economies, and more resilient communities. Climate finance has increased by 10% since the previous year [2,7].

Although the regulatory quality was not included, numerous studies focused on identifying the determinants of climate finance. For instance, [8–13] & [14] are some of them.

Contrary to this, linking climate finance was done even though regulatory quality and the issue of governance were hardly emphasized. For instance, [14–22], and [23] are a few of them. Expect the empirical work of [24], who included climate finance in modeling poverty reduction; however, there are limited studies, dealt with the effect of regulatory quality in this context. The inclusion of governance in the study of climate finance was conducted by [25]. However, only a few studies introduced regulatory qualities-climate finance interactive terms to investigate inclusive growth. This study investigates the linkage between climate finance and inclusive growth after the regulatory quality is introduced in the inclusive growth model of Africa.

This study is unique from the previous studies as it incorporates the interactive terms of regulatory quality-climate finance in Africa's inclusive growth model. In addition, it updates data for the study from 2013 to 2023. Finally, this study is also unique as it covers a relatively large sample (the 54 African countries). Concerning the scope interms of time, this study covers 11 years of study from 2013 to 2023, while it covers the entire continent of Africa in terms of geographical scope. Further, the study focuses on examining how regulatory quality affects the link between climate finance and inclusive growth in Africa.

The rest sections of the study are organized as follows. Section 2 discusses literature reviews, while methodology is covered in Section 3. Results and discussions are included in Section 4, while Section 5 is dedicated to discussing conclusions and recommendations.

## 2. Literature review

### 2.1. Regulatory quality, climate finance, and inclusive growth: Concept and bondage

Following the current volatile global dynamics, these three concepts, regulatory quality, climate finance, and inclusive growth, are currently at the top agenda of the world. Firstly, regulatory quality, in the light of governance quality, refers to how effectively governments design and implement rules that support economic activity [26]. The ability of the government to create and carry out sensible laws and regulations that support the growth of the private sector is gauged by the Regulatory Quality Index (RQI) [27]. It includes a number of metrics that highlight the substance and clarity of laws [6]. With values ranging from −2.5 (poor) to 2.5 (strong), the index is a component of the Worldwide Governance Indicators (WGI), which rates nations according to the strength of their regulations [28]. For example, Singapore had the highest score of 2.31 in 2023, while the average score was −0.03. According to [29,30], the indicator is essential for comprehending regulatory governance and how it affects economic performance [31,32].

Secondly, climate finance is funding from public and private sources to mitigate and adapt to climate change. Inclusive Growth (IG) is a broadly shared concept aiming at reducing poverty and inequality while preserving freedoms [33]. For instance, [34] emphasized that climate finance is the money that is targeted at the mitigation and adaptation of climate change to make the environment safe and comfortable for the benefit of the common [5,35]. The definition provided by the United Nations is "funding that tries to maintain and improve the resilience of biological and human systems to the adverse effects of climate change, as well as to reduce emissions and improve greenhouse gas sinks" [34]. Following this, the publication of [36] made the climate finance research very promising [37]. The recognition of the obvious advantages of environmentally friendly energy sources gave rise to the concept of climate change. It is closely linked to the renewed interest in renewable energy sources around the world [35]. The Paris Agreement is mostly linked to climate finance [12].

Thirdly, the ability of the government to enact and enforce reasonable laws and regulations that encourage the expansion of the private sector is known as regulatory quality [38,39]. According to [26,40], it includes elements like the government's ability to enforce regulations as well as their efficacy, efficiency, and clarity [41,42]. Encouraging a business-friendly and economically stimulating environment requires high-quality regulations [43]. The World Bank and OECD are among the institutions that use indicators to analyze the governance aspects of regulatory reforms in order to determine the quality of regulations [44].

For instance, improvements in life expectancy and infant mortality did not coincide with Great Britain's high during the Industrial Revolution [45]. Inclusive growth is defined as that which benefits everyone in society, especially those in low-income and vulnerable groups [46]. It highlights the significance of fostering social inclusion and lowering poverty in order to guarantee that everyone has equal access to opportunities [47]. Inclusive growth is frequently linked to pro-poor growth initiatives, which seek to improve human skills and generate productive jobs by investing in social services, health care, and education [40,48]. In order to address inequality and advance equity in economic development, this strategy links individual opportunities to macroeconomic determinants [49,50].

### 2.2. Theoretical base of the study

There are solid theoretical foundations for the relationship between regulatory and climate finance. Strong, transparent, and predictable regulatory regimes are crucial for mobilizing and successfully implementing climate finance, according to theories connecting regulatory quality with climate finance [51]. Inadequate regulation reduces the impact of public climate funds, discourages private investment, and causes uncertainty. The relationship between climate finance and regulatory quality is supported by a number of studies [26].

The first is the hypothesis of institutional quality, which claims that the efficiency of climate finance is based on institutional strength [52]. Furthermore, nations with strong regulations lower the likelihood of inefficiency, misallocation, and corruption. Additionally, this draws private investment and international climate money due to investors' confidence in the enforcement of regulations [50]. Endogenous growth theory and this hypothesis are closely related. In order to create an environment that promotes innovation and entrepreneurship, it highlights the significance of institutions, policies, and investments [53]. To put it another way, endogenous growth theory contends that growth results from both endogenous and external variables, both of which can be impacted by public policy [54].

Second, the relationship between climate funding and regulatory quality is based on the policy credibility theory. Another idea that introduced the foundation of climate finance and regulatory quality is this one [55]. According to this idea, consistent regulations indicate a sustained commitment to climate goals [56]. Furthermore, predictable policies reduce perceived risk (e.g., carbon pricing, renewable energy prices) [57]. Additionally, this trustworthiness promotes long-term investments and blended finance [58].

Thirdly, the theory that supports the problem is financial regulation and risk mitigation theory. According to this hypothesis, financial flows can be redirected toward low-carbon investments by regulators [59]. Furthermore, efficient regulation lowers political risk and policy uncertainty, increasing the bankability of climate initiatives [60]. The rationale behind and methods used by governments to create financial market regulations are explained by the theory of financial regulation. It primarily focuses on three strategies: self-regulation, systemic stability regulation, and command-and-control regulation [56].

Lastly, this study makes use of institutional quality theory, which is closely related to endogenous growth theory. Therefore, the role of climate finance and regulatory quality on inclusive growth is investigated in the present study. Further, the effect of two key variables, the interactive terms of regulatory quality and climate finance, was regressed on inclusive growth to examine their short and long term effects in Africa.

### 2.3. Studies on the mediating roles of regulatory quality on the link between climate finance and inclusive growth

Several empirical studies examined the linkage between regulatory quality and climate finance. For instance, [61] investigated how regulatory quality moderates the relationship between financial development and inclusive growth. The empirical work of [62,63] popularized the effect of governance indicators; regulatory quality is one of them, on climate finance. Similarly, [64] found that financial regulations can enhance the relationship between financial inclusion and growth. Another work of [65] emphasized that institutional quality affects the link between green finance and institutional qualities. Another empirical work that established the link between finance and regulatory quality is the work of [31]. From this literature, it is possible to understand that regulatory quality and the financial system, including climate finance, can be linked [66].

On the other hand, regulatory quality mediated the link between financial sector development and growth of information technology in the work of [67]. This shows that there are empirical grounds for the regulatory quality to mediate climate finance and inclusive growth. Similarly, [68] found that financial regulation mediates climate finance and renewable energy. Further, [69,70] revealed that climate finance regulation itself uniquely required contributing to climate finance. On the other hand, [71,72] also emphasized that climate finance can be regulated through climate policies. From all this, it is possible to conclude that regulatory quality can mediate the link between inclusive growth and climate finance. Similarly, [73] found that climate funds and the resilience of an economy can be mediated by sustainable investments, which are the function of regulatory quality. The work of [68] found that governance, which incorporates regulatory quality, can mediate the link between climate change in Asia. Another empirical work that viewed climate policy through the lens of governance is that of [74]. Their empirical work also underlined the moderating role of governance, which includes regulatory quality in several economic studies [75].

### 2.4. Empirical studies on the relationship between climate finance and inclusive growth

Several studies have been carried out regarding the effect of climate finance on environmental quality. For instance, firstly, [15] examined the effect of climate finance on inclusive growth using a panel propensity score matching model and found that

climate finance is essential for significantly reducing greenhouse gas emissions. Their study also found that climate finance supports energy sustainability. Their study mainly focuses on getting the distinction between the groups regarding this.

Secondly, [76] found that climate finance and international trade are good for the inclusive growth of Africa. Their study used 54 African countries and panel data from the period of 2004–2022. Their study further concluded that the climate finance effect can be increased when African countries trade with each other.

Thirdly, [77] found that climate finance enhances environmental quality in a positive and significant way. Using the panel of 111 countries around the globe from the period 2000–2019 and panel GMM, their study concluded that climate finance supports the betterment of environmental quality before the global millennium.

Fourthly, [16] found that the export robustly supports climate aid. Using a two-stage least squares for instrumental variables, their study found that the terms of trade of countries support climate aid positively from 2002–2017. Fifthly, [78] found that climate finance to mitigate climate risk is not evenly distributed across the globe. Using the panel data for the period of 2011–2021, the study found a negative linkage between climate finance and long-term climate risks, as South Asians are receiving a smaller amount of climate health due to the uneven distribution of climate health across the globe.

Sixthly, [79] studied the link between climate finance and developing countries and found that there is a strong and positive correlation between the two. They argued that climate finance solves the financial shortages of entrepreneurs in developing nations by promoting the supply of finance in such regions. Another study considered climate finance with poverty reduction and specifically with women's hunger alleviation [9,24].

Seventhly, [17] found that climate finance alleviates environmental degradation and improves human development by driving decarbonization. Using annual balanced panel data from 2001 to 2019 covering 36 developing economies, their study concluded that regulation quality decreases environmental degradation. Similarly, [18] found that climate finance suffers from several geographical injustices and world system dependencies. However, other research presents conflicting findings. First, [23] found that climate finance improves mitigation and adaptation measures, while [80] found that climate financing promotes corruption in developing nations.

Similarly, although it is statistically insignificant, the study by [81] found that climate finance had a favorable impact on mitigation. Furthermore, whereas an empirical study by [79] suggested that climate finance has a knock-on effect on entrepreneurship in underdeveloped nations, [77] demonstrated that climate financing improves environmental quality [82]. Eighthly, [19] found that there is a potential correlation between climate funding and the reduction of carbon emissions. Using the panel data from 1999 to 2017, their study concluded that climate finance enhances the development of a greener future. This fact is confirmed by similar studies such as [10].

Ninthly, [83] found that energy finance positively enhances. Using the panel data from the WDI from 2000 to 2023 for BRIC member countries, their study concludes that governance enhances the way energy finance improves the performance of the member countries [84]. Contrary to this, [85] found that climate finance is negatively contributing to carbon emissions from the period of 1990–2020 in G-7 economies. [86] found that green finance supports combined with sustainability, in G-20 countries from the period of 2010–2020, through the use of Quantile-on-Quantile regression [36].

Table 1 summarizes the previous empirical studies on the effect of climate finance on inclusive growth. Although the debate is hot in the current literature, the issue of how regulatory quality affects the relationship between climate finance and inclusive growth was not discussed in these empirical studies. This is the literature gap that this study aims to fill. The literature breach observed here is that there is a lack of consensus among the empirical studies on the link between climate finance. On the other hand, only a few studies have been conducted in Africa. The majority of the studies were conducted using the data of developing economies. This study conducts an examination of climate finance, regulatory quality, and inclusive growth using panel data of African countries from the period of 2013–2023.

**Table 1. Summary of empirical studies on the effect of climate finance, regulatory quality, and inclusive growth.**

| Authors | Type of data | Year included | Regions | No. of countries | Findings |
|---|---|---|---|---|---|
| [76] | Panel data | 2004-2022 | Africa | 54 | Positive linkage |
| [22] | Panel data | 1990–2015 | Developing countries | 141 | Positive effect |
| [77] | Panel data | 2000-2019 | Developed & developing countries | 111 | Positive linkage |
| [80] | Panel data | 2015–2021 | Developing countries | 74 | Positive linkage |
| [87] | Panel data | 2000–2021 | Developing countries | 130 | Causes corruption |
| [88] | Panel data | 2000–2022 | Developing countries | 117 | Positive linkage |
| [17] | Panel data | 2001–2019 | Developing countries | 36 | Positive |
| [23] | Panel data | 2010–2021 | Developing countries | 30 | positive |
| [79] | Panel data | 2007 −2018 | Developing countries | 101 | Positive |
| [14] | Panel data | 2011- 2021 | Developed & developing countries | 147 | Positive |
| [89] | Panel data | 1999 −2017 | Developing countries | 129 | Positive |
| [73] | Panel data | 1990 −2023 | Developing countries | 35 | Positive effect of climate fiancé |
| [90] | Panel data | 1990- 2021 | OECD economies | 19 | Positive effects of green finance |
| [91] | Panel data | 1995–2023 | Resource-rich countries | 15 | Green finance enhances innovation |
| [24] | Panel data | 2006–2017 | Selected SSA countries | 44 | Climate finance helps poverty alleviation |

Source: Author construction, 2025.

## 2.5. Conceptual framework

The conceptual framework of this study, as depicted in Fig 1, is developed based on the fact that it investigates the effects of climate finance on inclusive growth in Africa and examines whether regulatory quality moderates this relationship. The conceptual framework is created to reflect how regulatory quality affects the linkage between climate finance and economic growth.

## 2.6. Literature gap

Although plenty of studies were conducted on the effect of climate finance on growth, only a few studies included regulatory quality as a moderator. Further, this study explicitly emphasized regulatory quality, unlike other previous studies that emphasized governance. Further, this present work introduced the interactive term of regulatory quality-climate finance interaction to add to the few existing studies in the updated data and large cross-sections of African countries.

## 3. Research methodology

### 3.1. Data and variables

This section focuses on describing data and the variables of the study. The sources of each data are also well explained in this section. Accordingly, in this study, balanced panel data from the period of 2013–2023 is used for the 54 African countries. Table 2 displays the study variables employed to examine the effect of climate finance on the inclusive growth of Africa. Accordingly, the inclusive growth as measured by the inclusive growth index, 2021, is a dependent variable of the study, while climate finance is measured by climate-based net official development assistance (ODA) received [1]. The inclusive growth index is sourced from UNCTAD [30,92]. While all other explanatory variable data are obtained from the World Development Indicators [3]. Regulatory quality is captured in this inclusive growth model through its estimate following the World Bank database [2,4]. The estimation of the results for this study was conducted using STATA15.

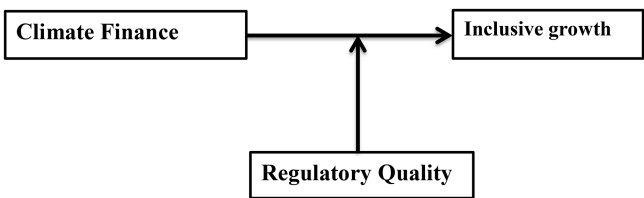

**Fig 1. Conceptual framework of the study, Source: Author's construction, 2025.**

**Table 2. Study variables and sources.**

| Study variables | Abbreviation | Source |
|---|---|---|
| Inclusive growth index,2021 (dependent variable of the study) | IGIN | UNCTAD,2025 |
| Net official development assistance received (constant 2021 US$) | CLIF | WDI,2025 |
| Regulatory Quality: Estimate | REGU | WDI,2025 |
| Population, total | POPL | WDI,2025 |
| Trade (% of GDP) | TRAD | WDI,2025 |
| Gross fixed capital formation (annual % growth) | GROF | WDI,2025 |
| External debt stocks (% of GNI) | DEBS | WDI,2025 |

**Note:** UNCTAD is the United Nations Conference on Trade and Development; WDI stands for World Development Indicator.

### 3.2. Climate-based ODA as a proxy of climate finance

Since official development assistance (ODA) provides the most appropriate and comparable data, it was employed in this study as a measure of climate funding [93]. Furthermore, it is becoming more common to use ODA as a proxy for climate finance [72,94]. Additionally, it offers a dataset that is consistent, comparable, and verifiable. Official development assistance (ODA) is also regarded as a useful and crucial indicator of climate funding. It was used in this study. Furthermore, as a fundamental component of global climate financing, it monitors Public Financing Commitments [95]. Government funds are the subject of the ODA pledge [96,97].

By definition, ODA is public funding intended for development. As a result, it's the easiest method to monitor the public aspect of these international commitments. Development and climate actions are closely related [22]. Numerous climate-related projects are also essential to development [8]. By avoiding soiled methods, ODA enables the integration of development and climate agendas [9]. Marking ODA flows as "climate-related" is an attempt to quantify this "additional" effort beyond typical development aid, although it is difficult in practice [76].

### 3.3. Methodological frameworks

This study employs the endogenous theory of growth that puts capital, labor, and technology at the center. Following the introduction of the theory, the link between climate funds and inclusive growth was previously modeled by several scholars [54,62]. For instance, [20] modeled inclusive growth as a function of inclusive finance, while [98] explained inclusive growth as a function of digital finance and financial inclusion. Further, [15] modeled it as a function of climate funds, whereas [21] explained that climate finance readiness is very crucial for sustainable development. Similarly, [99] modeled it as a function of green finance. Therefore, this study includes climate finance in the inclusive growth model following the empirical contributions of [15] and [76].

Accordingly, the inclusive growth model in this study is provided as:

$$IGIN = f(CLIF, REGU) \tag{1}$$

Where *CLIF* is climate finance as measured by the net official development assistance received, and *REGU* is the regulatory quality estimate by the World Development Indicator.

Equation 1 shows that inclusive growth in Africa is a function of climate finance and regulatory quality.

The model in which the interactive variable of the climate finance-regulatory quality is included is given as:

$$IGIN = f(CLIF, \ REGU, (CLIF * REGU)) \tag{2}$$

Where $(CLIF * REGU)$ shows the interactive term of the climate finance and regulatory quality of the model, and all others are explained above.

Equation 2 displays the introduction of the interactive term of the climate finance and regulatory quality in the inclusive growth model to show how regulatory quality enhances the impact of climate finance on inclusive growth. The interactive term is developed here to show how regulatory quality moderates the effect of climate finance on inclusive growth.

### 3.4. System GMM model specification

This study employed the dynamic generalized method of moments (GMM) as recommended by [100]. This method of estimation is preferred over the other as it allows flexibility in the identification of instruments [101]. Further, the generalized method of moments works better for large samples in the panel data. It also permits the number of moment conditions to be larger than the number of parameters, which is desirable in such empirical works [102].

This study modeled climate finance as a function of inclusive growth based on the empirical works of [15,103], and [77]. However, the current study improves on their work by including regulatory quality in the inclusive growth model to see its mediating role on the relationship between climate finance and, following the empirical works of [89], who modeled green growth as a function of regulatory quality. This study is better than the previous studies in that it incorporates both regulatory quality and climate finance in the inclusive growth model, with the main objective of examining the moderating effect of regulatory quality and climate finance on inclusive growth in Africa.

Thus, the functional form of the model, along with the control variables, is given as:

$$IGIN = \ f(CLIF, \ REGU, \ POPL, \ TRAD, \ GROF, \ DEBS, (CLIF * REGU)) \tag{3}$$

Where, *IGIN* includes growth as measured by the incline growth index by the United Nations, *CLIF* is climate finance, *REGU* is the regulatory quality estimate by the World Development Indicator, *POPL* is population. *TRAD* is trade of goods and services; *GROF* is gross fixed capital formation, and *DEBS* is external debt stock, and $(CLIF * REGU)$ is the interactive term of the regulatory quality and climate finance.

Equation 3 shows the functional form of the model in which inclusive growth is modeled as a function of climate finance and regulatory quality, as well as the control variables of the study.

The Generalized Method of Moments (GMM) model is provided as:

$$IGIN_{it} = \phi + \phi_1 IGIN_{it-1} + \phi_2 CLIF_{it} + \phi_3 REGU_{it} + \phi_4 POPL_{it} + \phi_5 TRAD_{it} + \phi_6 GROF_{it} + \phi_7 DEBS_{it}$$
$$+ \ \phi_8 (CLIF * REGU)_{it} + \psi_{it} + \varphi_{it} \tag{4}$$

Where, $IGIN_{it-1}$ is the first lag of the dependent variable of the study, all others explained in equations 1, 2, and 3 above, $\phi$s are the constant and coefficients of the explanatory variables respectively, *it* are cross sections (countries) and time

(years) of the study, while $+\psi_{it}$ is the unobserved country-specific effects of country $i$ at time $t$, and $\varphi_{it}$ Shows an independent and identically distributed error term in the model.

Equation 4 shows the specified GMM model before transformation, in which Africa's inclusive growth is modeled from climate finance and regulatory quality.

Finally, the transformed model of the study is provided as:

$$logIGIN_{it} = \phi + \phi_1 logIGIN_{it-1} + \phi_2 logCLIF_{it} + \phi_3 logREGU_{it} + \phi_4 logPOPL_{it} + \phi_5 logTRAD_{it} + \phi_6 logGROF_{it}$$
$$+ \phi_7 logDEBS_{it} + \phi_8 log(CLIF * REGU)_{it} + \psi_{it} + \varphi_{it} \tag{5}$$

Where $log$ is the natural logarithm of the format of the model, and all variables of the model are duly transformed.

Equation 5 shows the transformed model of the study to the natural logarithm. The purpose of transforming this model is to maintain a very clear interpretation of the model in the form of elasticity, which is desirable and intended to obtain by how much the inclusive growth of Africa changes for a percent change in climate finance and regulatory quality of Africa.

Another note here is how the logarithms on variables that contain zeros or negative values, such as regulatory quality, which ranges from −2.5 to +2.5, are solved. This study used a linear Transformation for Logarithmic Conversion for the regulatory quality(REGU) that contained negative values. This method of conversion is preferred as it is the simplest and most widely used method to enable logarithmic transformation of variables that contain negative values. The author added a constant to *every* observation to shift the entire distribution of the regularity quality into positive territory.

## 4. Results and discussions

### 4.1. Results of descriptive statistics of the study

Table 3 displays descriptive statistics of the study. Accordingly, the mean of inclusive growth of Africa is found to be 24.876, showing one of the lowest inclusive growth indexes in the world. According to UNCTAD, the average inclusive growth of the world is 40, while the average inclusive growth index of developing countries is 35. This shows that the average inclusive growth index of SSA countries is below the average of the developing countries' inclusive growth index. The UNCTAD Inclusive Growth Index evaluates living circumstances, equity, and environmental sustainability in addition to more conventional economic indices like GDP.

Similarly, the average climate finance received in Africa over the study period is found to be 8.75 trillion dollars, showing that Africa is one of the major recipients of climate finance assistance in the world. This is with the standard deviation of 2.24, implying that the climate finance assistance received is different across African countries.

**Table 3. Results of the descriptive statistics of the study.**

| Variables | IGIN | CLIF | REGU | POPL | TRAD | GROF | DEBS |
|---|---|---|---|---|---|---|---|
| Observations | 1,296 | 1,296 | 1,296 | 1,296 | 1,296 | 1,296 | 1,296 |
| Mean | 24.87692 | 8.7508 | −0.727384 | 2.09120 | 70.1956 | 10.59321 | 58.6298 |
| Standard deviation | 0.042652 | 2.2408 | 0.0472371 | 3.65626 | 5.68480 | 14.981 | 20.67502 |
| Min | 24.87692 | 4.3908 | −0.8019949 | 1.5400 | 59.0085 | −4.159147 | 33.51943 |
| Max | 24.87692 | 1.3009 | −0.6456561 | 2.7410 | 79.2559 | 78.21848 | 97.53926 |
| Skewness | 0.3307 | 0.3472 | 0.1160 | 0.7231 | 0.0215 | 0.1103 | 0.1049 |
| Kurtosis | 3.5645 | 4.8732 | 3.0237 | 2.2227 | 0.8311 | 0.1249 | 0.0399 |

**Note:** IGIN shows, CLIF is climate finance, REGU is regulatory quality, POPL is population, TRAD is trade of goods and services, while GROF is gross fixed capital formation, and DEBS is external debt stock. Prob>chi2 is greater than 0.05.

As far as the regulatory quality of Africa is concerned, it is found to have a mean value of negative (−0.727384), reflecting that Africa's regulatory quality is one of the lowest regulatory quality estimates. Further, the standard deviation of 0.0472371 depicts that the level of regulatory qualities varies across the countries of Africa.

Concerning the skewedness, the data has shown that it is a positively skewed distribution of inclusive growth. It shows that the difference between the tails of the data distribution is fatter than the other. All variables are found to be positively skewed. On the other hand, the kurtosis shows that inclusive growth, climate finance, and regulatory qualities have long tails with a sharp peak; heavy tails. The variables have heavy tails because their kurtosis is greater than three.

Table 4 displays the results of the Spearman correlation matrices of the study. Accordingly, the correlation between climate finance in Africa is found to be positive and strongly correlated to inclusive growth, with a coefficient of 0.8287. This result is found to be significant at 5% level of significance. This shows that the climate finance and inclusive growth nexus is worth studying, and the linkage between the two. On the other hand, the regulatory quality is found to have a strong positive correlation with the inclusive growth index of Africa, with a coefficient of 0.9443. This implies that although it has a low and negative estimate, the regulatory quality in Africa is positively correlated with inclusive growth.

### 4.2. Trends of Africa's climate finance(2013–2023)

Fig 2 depicts the trends of climate finance in Africa. It is shown that Africa's climate finance is showing rising trends. It follows that the official development assistance, the proxy of climate finance, is increasing, although there are fluctuations. From the year 2000–2007, the climate finance for Africa was rising, while it sharply declined from 2008 to 2010 due to the global financial crisis. Since then, although there is no sharp rise, the flow of official development assistance in the form of climate finance is rising in Africa.

Fig 3 shows the trends of regulatory quality in Africa. As we can see from this graph, the trend of Africa's regulatory quality was falling before 2005, implying that Africa was adversely impacted by the low regulatory quality during the first and second decades of the millennium. However, from the year 2005–2010, Africa's regulatory quality showed signs of improvement and was better, although it was not to the expected level. Finally, this figure also shows that a sharp fall in Africa's regulatory quality has been observed since 2019, associated with Covid19 pandemic. After the flood of COVID-19, Africa's climate finance remained low, although it indicated a sign of rising.

### 4.3. Panel unit root test results

Safety of the data is always the point of concern in empirical investigations. Thus, this study made three unit root tests to ensure that the balanced panel data is stationary. Moreover, the check of each unit root test was carried out at their level

Table 4. Spearman's Correlation test result.

| Correlation | IGIN | CLIF | REGU | POPL | TRAD | GROF | DEBS |
|---|---|---|---|---|---|---|---|
| IGIN | 1.0000 | | | | | | |
| CLIF | 0.8287* | 1.0000 | | | | | |
| REGU | 0.9443* | 0.9591* | 1.0000 | | | | |
| POPL | 0.4974* | −0.6255* | −0.8252* | 1.0000 | | | |
| TRAD | −0.500* | −0.1946* | −0.2913* | 0.6235* | 1.0000 | | |
| GROF | −0.2339* | 0.3614* | 0.4930* | −0.3774* | 0.1243* | 1.0000 | |
| DEBS | −0.0764* | −0.0069* | 0.0583* | −0.3504* | −0.7226* | −0.1478* | 1.0000 |

**Note:** * shows significance level at 5%, IGIN shows, CLIF is climate finance, REGU is regulatory quality, POPL is population, TRAD is trade of goods and services, while GROF is gross fixed capital formation, and DEBS is external debt stock.

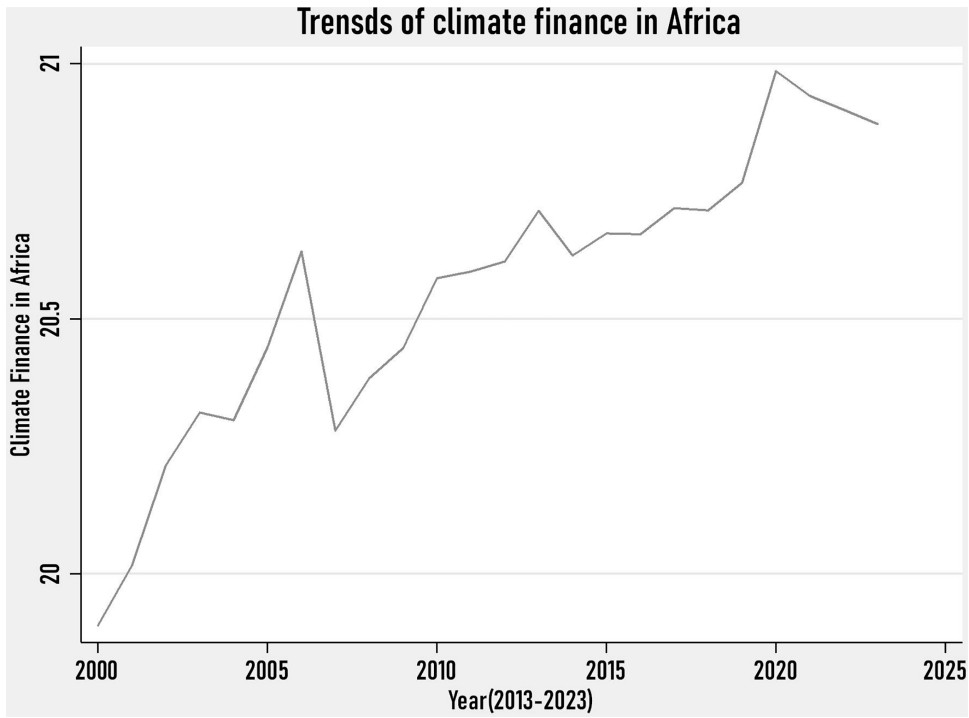

**Fig 2. Trends of Climate Finance in Africa.**

and first differences to make sure that the random walk of the data is not an issue that affects the quality of the study's findings.

The results of the panel unit root tests used to verify the safety of the balanced panel data are displayed in Table 5. As a result, three unit root tests were used for both first differences and levels. Thus, all variables are free from unit root at first difference, according to the Levin–Lin–Chu (LLC) unit root test result. The Harris-Tzavalis unit root test confirms the Im-Pesaran-Shin (IPS) unit root test's finding that all variables are stationary at the level. As a result, the three unit root tests adequately verified that there isn't a unit root issue. This illustrates how secure and clean the panel data is. After that, the data side granted authorization to run the system GMM model.

### 4.4. Cross-sectional dependence test results

The data should match as closely as feasible in any panel data analysis, especially in large cross sections (N) and brief periods (T). Thus, before doing the unit root test, the cross-sectional research test was conducted in compliance with Pesaran et al. (2008).

The results of the CD test employing the Friedman, Frees, and Pesaran tests are displayed in Table 6. The conclusion is that the model is safe and that going to the unit root is allowed by science because the probability values are bigger than $P > 0.05$ and the residuals are not reliant on the cross sections, which is the desired data.

### 4.5. Slope homogeneity test results

To select the most effective estimating method, the study employed the slope homogeneity test, which is recommended in the empirical studies of Hashem & Yamagata (2008) and Blomquist & Westerlund (2013). Both delta and its adjusted value were reported in this study.

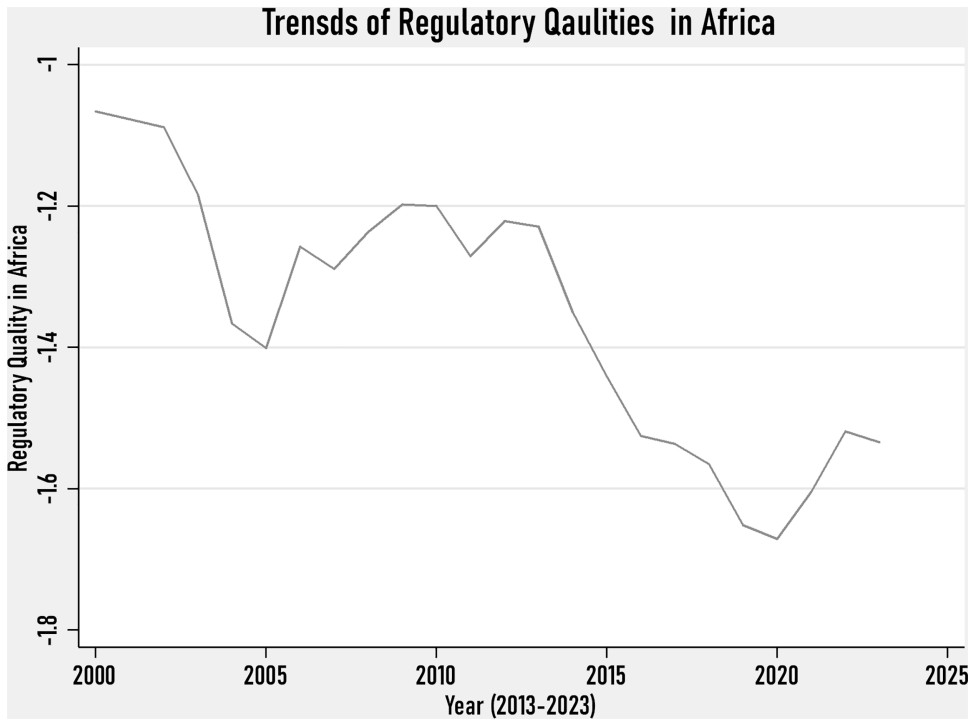

**Fig 3. Trends of Regulatory Quality in Africa.**

**Table 5. The result of panel unit root tests.**

| Variables | Methods of the unit root test | | | | | |
|---|---|---|---|---|---|---|
| | Levin-Lin –Chu | | Im–Pesaran–Shin | | Harris-Tzavalis | |
| | Level | First Difference | Level | First Difference | Level | First Difference |
| IGIN | −2.3697* | −4.5778** | −3.4424* | −6.9114* | −2.9936** | −3.7480 * |
| CLIF | −13.4037* | −8.0889 * | −14.8836** | −21.2215** | −1.4974* | −0.2693 * |
| REGU | −9.7196* | −19.3824 * | −2.9110* | −13.5232 | 0.1686* | 0.7157 |
| POPL | −20.7365* | −15.2240* | −0.5754* | −6.9114** | 10.4584* | 0.9674* |
| TRAD | −11.5223 * | −31.3402 * | −7.2452** | −17.9847* | −9.8540* | −29.6406* |
| GROF | −27.8817* | −43.8014* | −25.2139* | −9.7328** | −25.9130* | −26.2394* |
| DEBS | −16.5382 | −15.1056* | −0.6437** | −8.0422 ** | 9.7985* | 0.2617* |

**Note:** * and ** show significant level at 1% and 5% respectively. IGIN shows, CLIF is climate finance, REGU is regulatory quality, POPL is population, TRAD is trade of goods and services, while GROF is gross fixed capital formation, and DEBS is external debt stock.

Table 7 displays the slope homogeneity test results. Slopes are hence non-heterogeneous. This proves that N > T, and in this case, dynamic panel data—such as GMM—is superior to heterogeneous panel data.

### 4.6. Hausman (model selection) test results

In order to determine whether our data best fit with system GMM or difference GMM, our study used the Hausman test to regress climate finance over inclusive growth. The Hausman test is better than other methods for selecting between systems and various GMMs, according to Burgess & Harmon's (1991) research.

**Table 6. Cross-sectional dependence test results.**

| CD Tests | Commands | Statistics | Conclusion |
|---|---|---|---|
| Pesaran's test of CD | xtcsd, pesaran | 185.321* | Free from CD |
| Friedman test for CD | xtcsd, friedman | 1242.000* | Free from CD |
| Free tests for CD | xtcsd, frees | 51.652, alpha=0.05=0.1408 | Free from CD |

**Note:** CD is cross-sectional dependent, * P>0.05.

**Table 7. Results of slope homogeneity tests.**

| Slope Homogeneity Test Methods | Statistics | Commands | Conclusions |
|---|---|---|---|
| Pesaran, Yamagata. 2008. JE | 9.000*, 10.115 * | Xthst, Xs | Homogeneous slopes |
| Blomquist, Westerlund. 2013. EL | 11.61*, 13.805* | Xthst, Xs, hac | homogeneous slopes |

**Note:** JE is the journal of Econometrics, while EL is Economic Letters * P>0.05, Xs shows explanatory variables of the study.

The outcome of the Hausman model selection is shown in Table 8. After the test, the estimated result of the fixed effect model (0.0854) is less than the coefficient of lag-one of the dependent variable (L1. IGIN) computed using pooled OLS (0.0983). Additionally, the two-step difference GMM yielded the lowest estimate (0.0552). This suggests that, in comparison to the difference GMM, this specific data fits the system GMM the best. In order to investigate the immediate and long-term impacts of government spending and government digitization, this study used the system GMM estimating technique.

## 4.7. Effect of climate finance and regulatory quality on inclusive growth in Africa in the short run

This section is dedicated to discussing the short-run effect of climate finance and regulatory quality on the inclusive growth of Africa from the study period of 2013–2023. Accordingly, the two-step system GMM estimated is indicated as follows.

The short-term impact of climate finance and regulatory qualities on inclusive growth in Africa over the previous eleven years (2013–2023) is shown in Table 9. The system GMM result also shows that Africa's climate finance (logCLIF) is contributing positively to the continent's inclusive growth. The model's estimated conclusion showed that, when all other parameters are held constant, a 1% increase in climate funding is linked to a 0.156% rise in inclusive growth in Africa before short-term interactions with regulatory features. At the 1% level of significance, this result is considered significant.

This demonstrates how climate-official development assistance, which serves as a stand-in for climate finance in this study, is favorably influencing the inclusive growth of African nations. This suggests that the climate finance received is helping Africa. In contrast to [87]'s conclusion that climate finance contributes to corruption in developing nations, this result is consistent with the findings of [77], who showed that climate finance promotes international commerce and inclusive growth.

**Table 8. The result of the Hausman test of model selection.**

| Tests carried out | L1.logIGIN |
|---|---|
| POLS(Pooled OLS model) | 0.0983 |
| FEM(Fixed Effect Model) | 0.0854 |
| DGMM(Difference GMM model-two-step) | 0.0552 |
| Final decision | System GMM is more appropriate. |

**Note**: L1.logIGIN denotes the first lag of Inclusive growth, the dependent variable of the study.

**Table 9. Short-run effect of climate finance and regulatory quality on inclusive growth in Africa.**

| Variables | Two-step System GMM (A) | Z-value | System GMM model diagnosis |
|---|---|---|---|
| L1.IGIN | 0.5513856* (0.0730) | 1.541 | |
| logCLIF | 0.1561338*(0.0000) | 4.612 | Number of groups = 54 |
| Log(CLIF*REGU) | −0.7538109* (0.0000) | 1.813 | Number of instruments = 26 |
| logPOPL | −0.1906897*(0.0000) | 2.012 | Number of observations = 1296 |
| logTRAD | 0.7552188**(0.0000) | 3.012 | AR(1)=P > Z = 0.1831 |
| logGROF | 0.305914** (0.0000) | 2.214 | AR(2),P > Z = 0.5752 |
| logDEBS | 0.8012458*(0.0000) | 3.013 | Hansen test, P > Z = 0.2683 |

**Note:** in () are the robust standard errors, *,**, and *** show significance at 1%, 5%, and 10% level respectively, GMM is the generalized method of moments, AR(2) is Arrellano and Bond. IGIN shows, CLIF is climate finance, REGU is regulatory quality, POPL is population, TRAD is trade of goods and services, while GROF is gross fixed capital formation, and DEBS is external debt stock.

Another fascinating finding of the study is that the interactive term of the climate finance and regulatory quality (log-CLIF*REGU) adversely impacts the inclusive growth of Africa. The estimated result of the system GMM reveals that 1% increase in the interactive term of the climate finance and regulatory quality is associated with a 0.753% decline in inclusive growth of Africa, other things remaining constant. This indicates that although climate finance enhances inclusive growth before interacting with the regulatory quality, its positive impact is converted to negative due to the regulatory quality of Africa. This finding implies that the African government's capacity to create and carry out sensible policies and regulations is in doubt, which has a detrimental effect on the continent's climate financing contribution. Given that Africa has one of the lowest regulatory quality indices, this outcome appears to be sound.

According to the system GMM's findings, Africa's inclusive growth is being positively impacted in the short term by the first lag of inclusive growth (L1.IGIN). This variable's result suggests that the estimated system GMM model is accurate. Given that last year's inclusive growth was regarded as a key component of the economy, this outcome makes sense. Additionally, the economy presently benefits from having strong reserves from the past. According to the system GMM result, if all other factors remain unchanged, a 1% increase in inclusive growth from the previous year corresponds to a 0.551% rise in inclusive growth from this year in Africa. At the 1% level of significance, this result is deemed significant. This suggests a connection between the base's inclusive growth initiatives and the current year in Africa. This finding is limited to [103,104].

Further, the outcome of the study also shows that, when all other factors are maintained constant, a 1% rise in population growth is linked to a 0.190% short-term drop in inclusive growth. This suggests that a short-term population control measure is required in Africa. The study's findings support those of [105], who came to the conclusion that Africa's poor institutional quality negates the benefits of human capital development for the continent's inclusive growth.

The study's findings support those of [106,107], who came to the conclusion that Africa's poor institutional quality negates the benefits of human capital development for the continent's inclusive growth. This study's findings are limited to the research of [106], who suggested that a population check, including a check for unsustainable population growth, is necessary. [107] proposed using Africa's demographic dividend to reconcile this reality with inclusive growth, which this study also discovered [104].

Another finding from the system GMM is that trade of goods and services (logTRAD), which is positively enhancing the inclusive growth. It found that 1% increase in the trade of goods and services is associated with a 0.755% increase in inclusive growth of Africa in the short run, holding all other things constant. The result of this study is in line with [108], who found that trade positively contributes to the inclusive growth of SSA, and [108], who concluded

that agricultural trade encourages the inclusive growth of West African countries. Contrary to this, the findings of this study contradict the findings of [109], who concluded that trade liberalization is an evil to inclusive growth for West African countries.

The system GMM model estimated also revealed that gross fixed capital formation (logGROF) enhances inclusive growth of Africa. The result reveals that 1% increase in capital formation is associated with a 0.305% amplification in inclusive growth of Africa. This is a result found to be significant at 5% level of significance. This result contradicts the study of [110], who found that investment for inclusive growth leads to the loss of natural capital and, in turn, does not support inclusive growth as expected. However, this study agrees with the empirical work of [111], who confirmed the positive effect of capital formation on inclusive growth in Sub-Saharan Africa.

The study's findings also indicate that, in the short term, Africa's inclusive growth is being positively impacted by the external debt stock (logDEBS). When all other factors are held constant, the system GMM's projected conclusion shows that a 1% change in Africa's foreign debt stock corresponds to a 0.801% increase in the continent's inclusive growth. This outcome makes sense because the external debt stock is seen as a short-term source of funding since it stimulates the short-term financial supply. The empirical research of [111] and [112] is comparable to our study.

### 4.8. Model diagnosis and instrumentation mechanism

In the right-hand column of Table 9 is the model diagnosis, which shows the reliability of the model results. Accordingly, AR(2) is found to be 0.5752, insignificant, showing that the model is of the second-order serial correlation. This result is desirable as the model remains healthy when the second-order serial correlation is eliminated. Further, the result shows that the result of the model appears with AR(1)=P > Z = 0.1831, indicating that the problem of first-order serial correlation has nothing to do with this result. This implies that the model maintained the desirable AR(1) and AR(2). Similarly, the number of instruments is found to be 26 while the number of groups in the study is 54. This shows that the number of instruments is shortfalling the number of groups in the study, as the collapse option is used, indicating that the model is free from instrument proliferation, which is very undesirable. The fact that instrument proliferation is correct in the model simply depicts that the result of the model is reliable, as the correct instrumentation strategy is followed. Finally, the Hansen test of the results is found to be 0.2683, indicating that the instruments of the model are valid. Once the threats of instrumental validity are maintained, it follows that the results of the model are secured and the analysis, recommendations, and conclusions of the points are safe.

### 4.9. Effect of climate finance and regulatory quality on inclusive growth in Africa in the long run

This section discusses the long-run effect of climate finance and inclusive growth on inclusive growth in Africa. Table 10 shows the long-run effect of climate finance (logCLIF) on inclusive growth (logIGIN) in Africa. The result shows that Climate finance is increasing in Africa in the long run. The system GMM result reveals that a 1% change in official development assistance received in Africa is associated with a 0.3607263% increase in inclusive growth of Africa, on average, and other factors remained constant, in the long run. This is found to be significant at 1% level. This implies that donors need to continue to support climate finance as it promotes inclusive growth in the long run.

Contrary to this, Table 10 shows that when the climate finance interacts with the regulatory quality of the continent (Log(CLIF*REGU)), the positive effect of climate finance turns negative, showing that the available regulatory quality of Africa is adversely impacting the positive linkage between climate finance. Specifically, the result from the system GMM reveals that a 1% change in the interactive terms of climate finance and regulatory quality in Africa is associated with a 0.7750945% decrease in inclusive growth during the study period under consideration. The estimated equation also shows that the adverse effect of the regulatory quality is greater than the positive effect of climate finance. This shows that not only climate finance, but also strengthening regulatory quality in Africa needs special attention.

**Table 10. Long-run effect of climate finance and regulatory quality on inclusive growth of Africa.**

| Variables | Coefficient Results | t-value | P > |t| | Commands in STATA |
|---|---|---|---|---|
| L1.IGIN | 0.1061629 (.0308419) | 3.44 | 0.000 | nlcom (_b [L1. L1.IGIN])/ (1-_b [L1. L1.IGIN]) |
| logCLIF | 0.3607263 (.0140413) | 25.69 | 0.000 | nlcom (_b [logCLIF])/ (1-_ b [logCLIF]) |
| log(CLIF*REGU) | −0.7750945(.0053028) | −146.17 | 0.000 | nlcom (_b[Log(CLIF*REGU)})/(1-_b [Log(CLIF*REGU)]) |
| logPOPL | 0.9644879 (.0375428) | 25.69 | 0.000 | nlcom (_b[Log(logPOPL)})/(1-_b [Log(logPOPL)}) |
| logTRAD | 0.5117331 (.0601708) | 8.50 | 0.000 | nlcom (_b[Log(logTRAD)})/(1-_b [Log(logTRAD)}) |
| logGROF | −0.1033107 (.0043252) | −23.89 | 0.000 | nlcom (_b[Log(logGROF])/(1-_b [Log(logGROF)]) |
| logDEBS | −0.0513458(.0125678) | −4.09 | 0.000 | nlcom (_b[Log(logDEBS])/(1-_b [Log(logDEBS]) |

Note: IGIN shows, CLIF is climate finance, REGU is regulatory quality, POPL is population, TRAD is trade of goods and services, while GROF is gross fixed capital formation, and DEBS is external debt stock.

Furthermore, the first lag of inclusive growth (L1.IGIN) in Africa is found to be a positive and significant contributor in the long run. This shows that last year's accumulated inclusive growth is a base for the current year's inclusive growth, which looks logical. The estimated econometric result has shown that a 1% change in inclusive growth in Africa is associated with a 0.1061629% rise in inclusive growth of Africa. This reflects that a better foundation provides additional synergy to the economy.

Contrary to the short-term outcome, population growth (logPOPL) is found to be a substantial positive contributor to Africa's inclusive growth over the long term when considering the control variables. According to the system GMM, if all other factors stay the same, there is an average 0.9644879% improvement in inclusive growth in Africa for every 1% change in population growth. This is because the majority of the population is young people who enter the labor force on the continent. Further, the labor resource in Africa joins the labor market in the long run, and there is no threat that the loss of population will harm inclusive growth on the continent.

Comparably, it is discovered that Africa's trade in goods and services (logTRAD) significantly and favorably contributes to the continent's economic expansion. According to the system GMM model, a 1% shift in Africa's trade in goods and services is linked to a long-term rise in inclusive growth of 0.5117331%. This demonstrates how trade facilitates better resource distribution and boosts national economic expansion. Additionally, it improves foreign exchange availability, regional integration, and economic diplomacy, all of which contribute to the continent's inclusive prosperity.

On the other hand, it has been discovered that gross fixed capital formation (logGROF) in Africa eventually inhibits inclusive growth. According to the two-step method GMM, if all other factors stay the same, a 1% change in gross fixed capital formation in Africa is linked to an average long-term decline in inclusive growth of 0.1033107%. This demonstrates the need for better resource distribution in Africa.

In addition to this, the gross fixed capital formation in Africa is dominated by a few resource holders, while other sections of society are not. Finally, the external debt stock of Africa (logDEBS) is found to be a significant negative contributor to inclusive growth in the long run. The two-step system GMM estimated that a 1% increase in an external debt stock is associated with a 0.0513458% decrease in inclusive growth in the long run, on average, other things remaining fixed. This shows that the external debt stock in Africa is a burden on the generation in the long run. The further implication of this is that Africa needs to improve external debt stock management in the short run so that the long-run contribution turns positive through investment in profitable projects. This result conforms with the empirical findings of Anajama & Nyamudzanga (2024), who concluded that debt sustainability is a challenge in Africa.

## 5. Conclusions and recommendations

This study looked at how climate finance affected Africa's inclusive growth. It also looked into the short- and long-term effects of regulatory quality on the relationship between climate finance and inclusive growth in Africa. In order

to accomplish this goal, balanced panel data from the World Development Indicator and UNCTAD for the 2013–2023 research period were analyzed using a two-step system GMM (2SY-GMM).

The study concludes that climate finance enhances inclusive growth in both the short and the long run. Further, it's concluded that the positive effect of climate finance is larger in the long run compared to the short run. Before interacting with the regulatory quality, a one percent change in climate finance was enhanced by 0.3607 percent in the long run, while it was associated only with a 0.1561% rise in inclusive growth of Africa in the short run. This shows that climate finance contributes positively and significantly to the inclusive growth of Africa in both periods during the period under investigation. This implies that climate finance in Africa attracts special attention in both periods.

Contrary to this, in the interactive term of climate finance and regulatory quality of Africa (Log (CLIF*REGU) interact, the positive contribution of climate finance to inclusive growth turns negative. This implies that the level of regulatory quality in Africa adversely affects the inclusive growth of Africa in both the short and the long run. This shows that the effect of climate finance on inclusive growth in Africa is dependent on the level of Africa's regulatory quality. Another point of conclusion in this study is that the poor regulatory quality of Africa is as harmful in the long run as in the short run. Surprisingly, the system GMM model revealed that in the long run, a one percent change in the regulatory quality of Africa leads to a 0.775 percent decrease in inclusive growth, while it results in a 0.753 percent decrease in the short run, on average, and other things remaining constant. This reflects that improving regulatory qualities in Africa demands special attention from Africa's policymakers and development partners. Further, the policy implication of this study is that African governments need to reconsider the qualities of their economic policies, the capacity of law, regulations, and economic policy enforcement, efficiency, quality, and clarity of the regulations, and contextualize their economic policies with their corresponding business environments.

As far as the control variables are concerned, population was found to be a negative contributor to Africa's inclusive growth, while gross fixed capital formation, trade of goods and services, and external debt stock are found to be positive and significant contributors to inclusive growth in the short run. This implies that Africa needs to check population in the short run, whereas policies that enhance gross fixed capital formation and trade of goods and services need to be implemented by Africa's economic policymakers in the short run. Furthermore, the policy that improves external debt management and administration needs to be implemented by the policymakers of Africa in the short run to promote inclusive growth.

Contrary to this, in the long run, population growth and trade of goods and services in Africa are found to be a significant positive contributor to inclusive growth over the study period under consideration. This implies that long-run population checks are hardly recommendable in Africa as they hardly threaten inclusive growth. In addition to this, in Africa, policies that promote trade, trade integration, and the exchange of goods and services need to be implemented by Africa's policymakers.

Regarding gross fixed capital formation and external debt stock, the system GMM revealed that they are adverse and significant contributors to Africa's inclusive growth in the long run. If the inclusive growth of Africa is to be improved, the economic policies that enhance the distribution of gross fixed capital formation need to be implemented. The policies that promote investment in physical assets need to be revised and made more inclusive. The infrastructure and machinery investment law needs to be conducive in Africa if the inclusive growth of Africa is to be promoted in the long run. Further, the policies that limit external debt stock need to be implemented in Africa to enhance inclusive growth on the continent. It follows that the external debt stock accumulation in the long run needs to be discouraged in the long run.

### 5.1. Extolling further studies

The link between climate finance and inclusive growth is a burning issue of the 21st century. It is expected that this matter will be the discussion point of the literature. Thus, the next empirical studies are expected to focus on the role of institutional quality in mediating the link between institutional quality and inclusive growth, how governance affects the link between the two, and regional disparity issues across the continent.

## 5.2. Limitation of the study

Despite the advanced model, covered large cross-section and uses recent data, this study is not without limitations. This study covered only 11 years of data in terms of time, while it covers only African countries in terms of geography. Further, the study covered climate finance, which is measured by climate-based official development assistance. Further, regulatory quality is the only governance indicator used in the model. Further, this study carries all the limitations that system GMM has as an econometric model.

## Author contributions

**Data curation:** Isubalew Daba Ayana.

**Formal analysis:** Isubalew Daba Ayana.

**Investigation:** Isubalew Daba Ayana.

**Methodology:** Isubalew Daba Ayana.

**Project administration:** Isubalew Daba Ayana.

**Resources:** Isubalew Daba Ayana.

**Software:** Isubalew Daba Ayana.

**Supervision:** Isubalew Daba Ayana.

**Validation:** Isubalew Daba Ayana.

**Visualization:** Isubalew Daba Ayana.

**Writing – original draft:** Isubalew Daba Ayana.

**Writing – review & editing:** Isubalew Daba Ayana.

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
