## [Decision Letter · Decision Letter 0]

3 Dec 2025

PONE-D-25-47757Does the regulatory quality matter in the relationship between climate finance and inclusive growth in Africa?PLOS ONE?

Dear Dr. Ayana,

Thank you for submitting your manuscript to PLOS ONE. After careful consideration, we feel that it has merit but does not fully meet PLOS ONE’s publication criteria as it currently stands.

We agree with the reviewers that manuscript has potential but before to be considered some major concerns need to be addressed. We have concerns on the theoretical explanation of how regulatory quality moderates the relationship between climate finance and inclusive growth is not clearly developed. The literature review is unfocused and does not establish a convincing research gap or contribution. On the other hand, we find several methodological issues that can affect the reliability of the results. In addition, key variables are inconsistently defined, the rationale for focusing exclusively on Africa remains insufficient.

Given these concerns, we invite you to submit a revised version of the manuscript that addresses the points raised during the review process.

We look forward to receiving your revised manuscript.

Kind regards,

Juan E. Trinidad-Segovia, PhD

Section Editor

PLOS ONE

Journal Requirements:

Reviewers' comments:

Reviewer's Responses to Questions

**Comments to the Author**

1. Is the manuscript technically sound, and do the data support the conclusions?

Reviewer #1: Partly

Reviewer #2: Yes

Reviewer #3: Yes

2. Has the statistical analysis been performed appropriately and rigorously?

Reviewer #1: Yes

Reviewer #2: No

Reviewer #3: Yes

3. Have the authors made all data underlying the findings in their manuscript fully available?

Reviewer #1: Yes

Reviewer #2: No

Reviewer #3: Yes

4. Is the manuscript presented in an intelligible fashion and written in standard English?

Reviewer #1: Yes

Reviewer #2: Yes

Reviewer #3: Yes

Reviewer #1: The identified research gap lacks clarity, need improvement; the introduction repeatedly asserts that previous studies have not examined regulatory quality in the context of the climate-finance–inclusive-growth nexus. However, the literature review does not offer definitive systematic evidence demonstrating that regulatory quality has been entirely overlooked in governance-focused climate-finance research.

The study presents a degree of novelty; however, the emphasis on regulatory quality as a moderating variable is not fully substantiated. The theoretical rationale explaining the influence of regulatory quality on the transmission mechanism between climate finance and inclusive growth remains insufficiently articulated.

The contribution to the empirical literature appears to be overstated, especially given that many cited studies have already explored governance, institutional indicators, or policy quality in connection with climate finance or economic outcomes, which lessens the study's presentation of a completely new perspective. Author can improve the contribution part.

The relationship among climate finance, regulatory quality, and inclusive growth remains inadequately explored in the existing literature. The analysis examines these concepts in isolation rather than integrating them to illustrate the impact of regulatory quality on the equity-focused developmental results of climate finance.

The justification for focusing on Africa as a unique empirical context is not sufficiently compelling; author need to add more justification for inclusion of Africa

Minor:

Maintain consistency in words in some places Log is in capital and in some places it is small

Author can include a robustness estimate.

In the introduction author mentioned interactive variable abbreviation without explaining.

Author can enrich the literature by adding the following recent studies for improving methodology and literature review

The author can work on the policy recommendation and future research direction.

Does climate policy uncertainty abate financial inclusion? an empirical analysis through the lens of institutional quality and governance. Sustainability, 17(2), 520.

The moderating role of governance, banking regulation, and supervision on shadow economy, financial inclusion, and financial stability nexus: a case of G5 economies. Economic Change and Restructuring, 57(6), 176.

Reviewer #2: The manuscript investigates the effects of climate finance on inclusive growth in Africa and examines whether regulatory quality moderates this relationship. Using balanced panel data for 54 African countries from 2013–2023 and employing a two-step System GMM estimator, the study finds that climate finance positively influences inclusive growth in both the short and long run. Regulatory quality weakens and in fact reverses the positive effect of climate finance on inclusive growth. Several control variables (trade, capital formation, population, and external debt) exert varying short- and long-run effects. The study claims originality by incorporating a climate-finance–regulatory-quality interaction term in an inclusive growth framework.

While the study is topical and contextually relevant, the following issues need to be addressed before going forward.

1. The theoretical justification for why regulatory quality should moderate the climate-finance–inclusive-growth link is underdeveloped.

2. The literature review is lengthy but unfocused; it mixes descriptive information with unrelated empirical findings and lacks a conceptual framework.

3. Key constructs (inclusive growth, climate finance, regulatory quality) are described, but the connection among them is weak, and no mechanism is clearly articulated.

4. There seems to be serious issues with the data development. For example, the study uses Net Official Development Assistance received as a proxy for climate finance. This is not an accepted or valid measure unless proven with citations of other credible and relevant studies that have used it. ODA contains many unrelated components (health, education, peacekeeping, governance, social sectors). This undermines the validity of the findings and threatens internal consistency.

5. There are also methodological and econometric issues that need to be resolved

(a) Tables report extremely large or abnormal z-values (e.g., 4.6e+12), which suggest numerical instability, scaling problems, or incorrect specification.

(b) There is a potential for over-instrumentation risk in the model. Though the author notes 26 instruments for 54 groups, no discussion is provided on collapsing instruments or addressing instrument proliferation, which can weaken Hansen tests in System GMM.

(c) The paper incorrectly interprets IPS and LLC tests, reporting significance as “free from random walk,” which is not accurate. A clearer explanation is needed.

6. There are inconsistent variable definitions and misleading interpretation. For example, the dependent variable is “inclusive growth index (IGIN)”, yet the study keeps referring to it as “economic growth”. These are conceptually different variables.

7. Using logarithms on variables that contain zeros or negative values such as regulatory quality which ranges from −2.5 to +2.5 is mathematically invalid, unless adjusted.

8. Interaction term log(CLIF*REGU) is problematic due to negative REGU values.

9. Although the study claims originality by interacting term of climate financing and regulatory quality (CLIF*REGU). This is common in development economics literature. The manuscript should better articulate its contribution relative to: Institutional quality literature, climate finance governance and inclusive growth.

Reviewer #3: I commend the author for putting this manuscript together. However, a few issues abound.

1. The data and scope of the study was not well explained.

2. The literature review was scanty and for publication, the literature review approach should be at least a thematic analysis. The study is similar to

Doku, I. (2022). Are developing countries using climate funds for poverty alleviation? Evidence from Sub-Saharan Africa. The European Journal of Development Research, 34(6), 3026-3049. I will suggest that, the author should introduce something new.

**Do you want your identity to be public for this peer review?** For information about this choice, including consent withdrawal, please see our Privacy Policy

Reviewer #1: No

Reviewer #2: No

Reviewer #3: **Yes:** Isaac Doku

---

## [Author Response · Author response to Decision Letter 1]

19 Dec 2025

Responses to concerns from Reviewer #1:

1. The identified research gap lacks clarity and needs improvement; the introduction repeatedly asserts that previous studies have not examined regulatory quality in the context of the climate-finance–inclusive-growth nexus. However, the literature review does not offer definitive systematic evidence demonstrating that regulatory quality has been entirely overlooked in governance-focused climate-finance research.

Responses

Dear Reviewer #1, I would like to appreciate your suggestions and comments that improved the revised manuscript. Based on your comments, the clarity of the research gap is enhanced. A lot of improvements were made. The literature review section is improved by adding the research gap and the literature gap. Bade don your good comments, the revised manuscript has provided, the definitive evidence that the issue of regulatory quality is overlooked. ( see section 1 page 2 and 3 of the revised manuscript, Section 2 Page 11 of the revised manuscript).

2. The study presents a degree of novelty; however, the emphasis on regulatory quality as a moderating variable is not fully substantiated. The theoretical rationale explaining the influence of regulatory quality on the transmission mechanism between climate finance and inclusive growth remains insufficiently articulated.

Responses

Dear Reviewer, I would like to appreciate you once again for your nice comments. Accepting your comments positively and strongly, the revised manuscript has fully substantiated by the studies on regulatory quality, climate finance, and economic growth. Further, your comment enabled this manuscript to deserve sufficient articulation. (See section 2.3, 7 of the revised manuscript). Thank you.

3. The contribution to the empirical literature appears to be overstated, especially given that many cited studies have already explored governance, institutional indicators, or policy quality in connection with climate finance or economic outcomes, which lessens the study's presentation of a completely new perspective. Author can improve the contribution part.

Responses

Dear Reviewer, based on your comments, the over-stated contribution statement has been calmed down in the revised manuscript. The existing studies on the areas of regulatory quality, climate finance, and growth have gained the appropriate recognition. I would like to ask apology on this issue. Accordingly, the author improved the contribution part in the revised manuscript (section 1, see page 3, last paragraph of the revised manuscript).

4. The relationship among climate finance, regulatory quality, and inclusive growth remains inadequately explored in the existing literature. The analysis examines these concepts in isolation rather than integrating them to illustrate the impact of regulatory quality on the equity-focused developmental results of climate finance.

The justification for focusing on Africa as a unique empirical context is not sufficiently compelling; author need to add more justification for inclusion of Africa.

Responses

Dear reviewer, thank you so much for raising such an essential suggestion to enhance the revised manuscript. Based on your good comments, the revised manuscript has addressed all of your concerns on this. Accordingly, sufficient literatures were reviewed. This is done following your comments. The justification of the study is also revised in the revised manuscript. (see Section1 page 2 and 3, Section 2, pages 2.2 and 2.3, page 6 of the revised manuscript).

5. Minor:

a. Maintain consistency in words in some places, Log is in capital, and in some places it is small

Responses

Dear reviewer, this is done following your good comment. Proofreading concerning the Log and other related issues has been solved. The author conducted a careful proofreading of the manuscripts.

b. Author can include a robustness estimate.

Responses

Dear reviewer, the revised manuscript included robustness estimate in the result analysis. The model diagnoses section is included in section 4.8, page 24 of the revised manuscript. In addition, all standard errors of the model are estimated using robust standard errors.

c. In the introduction author mentioned an interactive variable abbreviation without explaining. Author can enrich the literature by adding the following recent studies for improving methodology and literature review.

Responses

Dear reviewer, I would like to appreciate your suggestion very timely and recent, to enhance the methodology and literature section of the revised manuscripts.

Based on your comments, two of the studies, the revised manuscript cited the following studies. (see the reference section). Thank you once again.

• Does climate policy uncertainty abate financial inclusion? An empirical analysis through the lens of institutional quality and governance. Sustainability, 17(2), 520.

• The moderating role of governance, banking regulation, and supervision on shadow economy, financial inclusion, and financial stability nexus: a case of G5 economies. Economic Change and Restructuring, 57(6), 176.

d. The author can work on the policy recommendation and future research direction.

Responses

Dear reviewer, based on your comments, the revised manuscript has improved the policy recommendation and future research. In addition to this, the revised manuscript has incorporated the limitations of the study. (See pages 28, 29, and 30 of the revised manuscript).

End of responses to Reviewer #1

Responses to concerns from Reviewer #2

Reviewer #2: The manuscript investigates the effects of climate finance on inclusive growth in Africa and examines whether regulatory quality moderates this relationship. Using balanced panel data for 54 African countries from 2013–2023 and employing a two-step System GMM estimator, the study finds that climate finance positively influences inclusive growth in both the short and long run. Regulatory quality weakens and in fact reverses the positive effect of climate finance on inclusive growth. Several control variables (trade, capital formation, population, and external debt) exert varying short- and long-run effects. The study claims originality by incorporating a climate-finance–regulatory-quality interaction term in an inclusive growth framework. While the study is topical and contextually relevant, the following issues need to be addressed before going forward.

Responses

Dear reviewer, thank you so much for understanding the manuscript. Exactly, the study is what you shortly articulated. Thank you for recommending that the study be topical and contextually relevant. Your understanding of cross section of the study (54) and the time of the study was amazing and correct. The model and the finding you summarized is also right. Thank you so much!

1. The theoretical justification for why regulatory quality should moderate the climate-finance–inclusive-growth link is underdeveloped.

Response

Dear Reviewer #2, thank you so much for providing very imperative suggestions to improve the manuscript. Based on your comments, the revised manuscript has organized the literature review section, emphasizing the theoretical grounds of the study. In addition to this, the literature review section was well organized in a thematic way. (Sections 2.2 & 2.3 page 5 and 6 of the revised manuscript). The Literature review section is well rewritten and organized in a more scientific way.

2. The literature review is lengthy but unfocused; it mixes descriptive information with unrelated empirical findings and lacks a conceptual framework.

Responses

Dear reviewer, I would like to appreciate you for raising these essential questions. Based on your good comments, the literature review is now well arranged and synthesized carefully (see pages 5 and 6 of the revised manuscript). Further, Table 1 on page 10 of the revised manuscript has synthesized the literature. Further, the conceptual framework is developed in the revised manuscript. ( see page 11 of the revised manuscript). Further, the gap in the literature is well identified. (see page 11 of the revised manuscript). Thank you once again.

3. Key constructs (inclusive growth, climate finance, regulatory quality) are described, but the connection among them is weak, and no mechanism is clearly articulated.

Responses

Dear reviewer, I would like to appreciate you for your nice suggestions. Based on your good comments, the theoretical grounds and the connections between the constructs are well organized. (See sections 2.2 and 2.3, page 5&6 of the revised manuscript).

4. There seems to be serious issues with the data development. For example, the study uses Net Official Development Assistance received as a proxy for climate finance. This is not an accepted or valid measure unless proven with citations of other credible and relevant studies that have used it. ODA contains many unrelated components (health, education, peacekeeping, governance, social sectors). This undermines the validity of the findings and threatens internal consistency.

Responses

Dear reviewer, this is a very critical suggestion. Thank you for rising. Accepting your comments positively, the revised manuscript has developed the section that supports the use of climate-based ODA as the measure of climate finance. (see section 3.2, page 11 of the revised manuscript). Thank you!

5. There are also methodological and econometric issues that need to be resolved

(a) Tables report extremely large or abnormal z-values (e.g., 4.6e+12), which suggest numerical instability, scaling problems, or incorrect specification.

Response

Dear reviewer, based on your good comments, the revised manuscript has addressed issues related to z-values. In addition to this, the revised manuscript revisited the methodology and corrected all issues around your comments. Thank you once again (see Table 10, page 26 of the revised manuscript).

(b) There is a potential for over-instrumentation risk in the model. Though the author notes 26 instruments for 54 groups, no discussion is provided on collapsing instruments or addressing instrument proliferation, which can weaken Hansen tests in System GMM.

Responses

Dear reviewer, thank you so much for your golden suggestion. Based on your comment, the revised manuscript has introduced the instrumentation strategy. The issue of instrument proliferation and the collapse option are used to eliminate the problem. This shows that the model employed in the study is free from the risk of instrumentation. ( see section 4.8, page 25 of the revised manuscript).

(c) The paper incorrectly interprets IPS and LLC tests, reporting significance as “free from random walk,” which is not accurate. A clearer explanation is needed.

Response

Dear Reviewer, thank you once again for the comment that is basic for the analysis. Based on your good comments, in the revised manuscript, the interpretation of unit root tests is corrected accordingly (see page 19 of the revised manuscript, all is highlighted).

6. There are inconsistent variable definitions and misleading interpretations. For example, the dependent variable is “inclusive growth index (IGIN)”, yet the study keeps referring to it as “economic growth”. These are conceptually different variables.

Responses

Dear reviewer, thank you so much for your suggestion that improved this manuscript. Based on your comment, the revised manuscript has improved all issues of inconsistency related to the variables. As you said, the dependent variable of the study is inclusive growth (IGIN), which is measured by the inclusive growth index. Thank you once again.

7. Using logarithms on variables that contain zeros or negative values such as regulatory quality which ranges from −2.5 to +2.5 is mathematically invalid, unless adjusted.

Reponses

Dear reviewer, this is another nice question that improved the current study. Based on your good comments, the revised manuscript has included the paragraph that explains the transformation of the values that contain negative numbers. (See page 15 of the revised manuscript.

8. Interaction term log(CLIF*REGU) is problematic due to negative REGU values.

Responses

Dear reviewer, this issue is solved as the linear transformation strategy is used ( as seen on page 15 ) of the revised manuscript. Thank you so much!

9. Although the study claims originality by interacting the term of climate financing and regulatory quality (CLIF*REGU). This is common in development economics literature. The manuscript should better articulate its contribution relative to: Institutional quality literature, climate finance governance, and inclusive growth.

Responses

Dear reviewer, thank you so much for raising this question. As you said, the issue of climate financing is common in economic literature. However, this study emphasized the regulatory quality component of governance, which only a few of the existing literatures considered as a moderator between the climate finance and inclusive growth. This is especially relevant in Africa, where the quality of the regulation is questionable. On top of that, in Africa, there are no sufficient studies that introduced regulatory quality in Africa’s inclusive growth model. Thus, this study adds some positive drop to the existing literature on this matter.

End of responses to Reviewer #2

Responses to concerns from Reviewer #3

Reviewer #3: I commend the author for putting this manuscript together. However, a few issues abound.

Responses

Dear Reviewer #3, thank you for providing me with an opportunity to organize the study. Based on your good comments, the revised manuscript has restructured the literature sections.

1. The data and scope of the study was not well explained.

Responses

Dear reviewer, thank you so much for providing me with very essential input that improved the paper. Based on your good comments, the data of the study is well elaborated. (See page 11 of the revised manuscript. Further, the revised manuscript included a paragraph that explains the scope of the study in the introduction section of the manuscript. (see page 3, the last highlighted paragraph in yellow.

2. The literature review was scanty and for publication, the literature review approach should be at least a thematic analysis.

Responses

Dear Reviewer, thank you so much for raising these great suggestions. Three of your work was cited in this manuscript. I would like to appreciate you for your wonderful contribution to the advancement of science in climate finance. Frankly speaking, I am proud to be with you in the world of research. Your three works cited in the revised manuscript are:

Doku, I., Richardson, T. E., & Essah, N. K. (2022). Bilateral climate finance and food security in developing countries: A look at German donations to Sub‐Saharan Africa. Food and Energy Security, 11(3). https://doi.org/10.1002/fes3.412

Doku, I., & Phiri, A. (2024). Climate finance and women-hunger alleviation in the global south: Is the Sub-Saharan Africa case any different? PloS One, 19(2), e0290274. https://doi.org/10.1371/journal.pone.0290274

Doku, I., Ncwadi, R., & Phiri, A. (2021). Determinants of climate finance: Analysis of recipient characteristics in Sub-Sahara Africa. Cogent Economics & Finance, 9(1). https://doi.org/10.1080/23322039.2021.1964212

3. The study is similar to Doku, I. (2022). Are developing countries using climate funds for poverty alleviation? Evidence from Sub-Saharan Africa. The European Journal of Development Research, 34(6), 3026-3049. I will suggest that, the author should introduce something new.

Responses

Dear reviewer, thank you for raising this question and suggestion. It is an imperative question. I would like to also appreciate your high-quality paper on this matter. For clarity, I would like to elaborate on the basic differences between Doku, I. (2022), and this current study.

Table 1: The distinction between Doku (2022) and the current study

Aspects of difference Doku (2022) This study

Data-wise difference 2006–2017 2013-2023

Country coverage 44 Sub-Saharan Africa 54 African countries

Dependent variable GDP per capita and social inequality. Inclusive growth

Mediating variables regulatory quality & control

---

## [Decision Letter · Decision Letter 1]

18 Jan 2026

PONE-D-25-47757R1Does the regulatory quality matter in the relationship between climate finance and inclusive growth in Africa?PLOS One?

Dear Dr. Ayana,

Thank you for submitting your manuscript to PLOS ONE. After careful consideration, we feel that it has merit but does not fully meet PLOS ONE’s publication criteria as it currently stands. Therefore, we invite you to submit a revised version of the manuscript that addresses the points raised during the review process.

We look forward to receiving your revised manuscript.

Kind regards,

Juan E. Trinidad-Segovia, PhD

Section Editor

PLOS One

Journal Requirements:

Reviewers' comments:

Reviewer's Responses to Questions

**Comments to the Author**

Reviewer #1: All comments have been addressed

Reviewer #2: All comments have been addressed

Reviewer #3: (No Response)

2. Is the manuscript technically sound, and do the data support the conclusions?

Reviewer #1: Yes

Reviewer #2: Yes

Reviewer #3: Partly

3. Has the statistical analysis been performed appropriately and rigorously?

Reviewer #1: Yes

Reviewer #2: Yes

Reviewer #3: Yes

4. Have the authors made all data underlying the findings in their manuscript fully available?

Reviewer #1: Yes

Reviewer #2: No

Reviewer #3: Yes

5. Is the manuscript presented in an intelligible fashion and written in standard English?

Reviewer #1: Yes

Reviewer #2: Yes

Reviewer #3: Yes

Reviewer #1: I am satisfied with the revised version and recommend acceptance in its current form.

Best Wishes.

Reviewer #2: Thank you for considering my comments in improving the quality of your manuscript. Continue improving on the manuscript until it fully meets the journal standard.

Reviewer #3: The authors did their best to address most of the issues raised, however a few things needed to be relooked at.

1. In the abstract, the authors mentioned that a 1 percentage change will lead to an increase in inclusive growth by 0.3607 percent. CWill that be the case if climate finance reduces? Since they kept using change.

2. The work needs serious editorial attention, I will suggest they give the work to a professional editor to work on. For instance, under literature review, they kept writing sentences like; "inclusive growth is that is, ". On page 9, they wrote " Table 1 shows the effect of climate finance on inclusive growth is the major point of debate among Although the debate is hot in the current literature". I think part of the sentences were cut away in generating the pdf version too.

3. The GMM results didn't mention what the values in bracket represents, but I presume they are standard errors.

**Do you want your identity to be public for this peer review?** For information about this choice, including consent withdrawal, please see our Privacy Policy

Reviewer #1: No

Reviewer #2: No

Reviewer #3: No

---

## [Author Response · Author response to Decision Letter 2]

21 Jan 2026

Responses to concerns from Reviewers

Reviewer #1: I am satisfied with the revised version and recommend acceptance in its current form. Best Wishes.

Response

Dear Reviewer #1, I truly appreciate your positive assessment of the revised version and thank you for recommending the manuscript for acceptance in its current form. I would also like to express my sincere gratitude to the reviewer for the valuable insights and constructive feedback provided during the review process. Your comments greatly contributed to improving the clarity and overall quality of the manuscript.

Reviewer #2: Thank you for considering my comments in improving the quality of your manuscript. Continue improving on the manuscript until it fully meets the journal standard.

Response

Dear Reviewer #2, thank you for acknowledging our efforts in addressing your comments and recommending my work for acceptance of my work in the venerated Plos One journal. I also sincerely appreciate your thoughtful comments and constructive feedback, which have greatly contributed to improving the overall quality of this manuscript. Your insights have been invaluable throughout the revision process, and I am grateful for the time and expertise you dedicated to reviewing my work. Finally, let me promise you that I will remain committed to continuing improvements to ensure that the manuscript fully meets the standards of the journal.

End of Responses to Reviewer #1 and #2

Reviewer #3: The authors did their best to address most of the issues rose; however a few things needed to be relooked at.

1. In the abstract, the authors mentioned that a 1 percentage change will lead to an increase in inclusive growth by 0.3607 percent. Will that be the case if climate finance reduces? Since they kept using change.

Responses

Dear Reviewer #3, thank you for rising very interesting suggestion regarding the interpretation of the coefficients in the introduction section. Based on your good comments, the revised manuscript has captured your concerns and the sentences in the abstract section are correctly interpreted. This is highlighted it in yellow. The opposite might not be true based on the system GMM estimation. Thank you once again (see line 7 of the abstract in the revised manuscript).

2. The work needs serious editorial attention; I will suggest they give the work to a professional editor to work on. For instance, under literature review, they kept writing sentences like; "inclusive growth is that is, ". On page 9, they wrote "Table 1 shows the effect of climate finance on inclusive growth is the major point of debate among although the debate is hot in the current literature". I think part of the sentences was cut away in generating the pdf. version too.

Responses

Dear Reviewer #3, thanks you so much, for your suggestion. Based on your good comments, the revised manuscript is carefully edited and proof read has happened to improve readability and clarity (see section 2 pages 3 and 4 of the revised manuscript, highlighted yellow for instance). Thank you.

3. The GMM results didn't mention what the values in bracket represent, but I presume they are standard errors. Reviewers

Responses

Dear reviewer #3, thanks you so much for your very professional insight to make the manuscript high quality paper. Based on your good comments, values in bracket represents are explained as standard errors just overlapping your presumption. As a scientific writer, I would like to apologise for such silly mistakes (See Table 9, page 22 of the revised manuscript).

End of Responses to Reviewer #3

I would like to appreciate all anonymous reviewers and the editor.

---

## [Decision Letter · Decision Letter 2]

15 Feb 2026

Does the regulatory quality matter in the relationship between climate finance and inclusive growth in Africa?

PONE-D-25-47757R2

Dear Dr. Ayana,

We’re pleased to inform you that your manuscript has been judged scientifically suitable for publication and will be formally accepted for publication once it meets all outstanding technical requirements.

Kind regards,

Juan E. Trinidad-Segovia, PhD

Section Editor

PLOS One

Additional Editor Comments (optional):

Reviewers' comments:

Reviewer's Responses to Questions

**Comments to the Author**

Reviewer #3: All comments have been addressed

2. Is the manuscript technically sound, and do the data support the conclusions?

Reviewer #3: Yes

3. Has the statistical analysis been performed appropriately and rigorously?

Reviewer #3: Yes

4. Have the authors made all data underlying the findings in their manuscript fully available?

Reviewer #3: Yes

5. Is the manuscript presented in an intelligible fashion and written in standard English?

Reviewer #3: Yes

Reviewer #3: Author has addressed all the comments provided and i wish to commend them for the hardwork.

I am ok with the manuscript in its current form and can be prepared for publication. Thanks

**Do you want your identity to be public for this peer review?** For information about this choice, including consent withdrawal, please see our Privacy Policy

Reviewer #3: No

---

## [Editor Report · Acceptance letter]

PONE-D-25-47757R2

PLOS One

Dear Dr. Ayana,

I'm pleased to inform you that your manuscript has been deemed suitable for publication in PLOS One. Congratulations! Your manuscript is now being handed over to our production team.

Kind regards,

on behalf of

Dr. Juan E. Trinidad-Segovia

Section Editor

PLOS One